# Liquid Foam-Ethyl Vinyl Acetate Adhesive Systems for Lining Process of Paintings: Prospects of a User-Friendly, Harmless Alternative to Conventional Products

**DOI:** 10.3390/polym15071741

**Published:** 2023-03-31

**Authors:** Gaia Tarantola, Elena Medri, Arianna Splendore, Francesca Lo Russo, Chiara Matteucci, Matteo Minelli

**Affiliations:** 1Department of Cultural Heritage, Alma Mater Studiorum—University of Bologna; I-48121 Ravenna, Italy; gaia.tarantola2@unibo.it (G.T.); chiara.matteucci@unibo.it (C.M.); 2Department of Civil, Chemical, Environmental and Materials Engineering, Alma Mater Studiorum—University of Bologna, I-40131 Bologna, Italy; elena.medri5@unibo.it; 3AF The Lab, Conservazione Restauro, I-20137 Milano, Italy; splendore77@me.com (A.S.); splendore.lorusso@gmail.com (F.L.R.)

**Keywords:** EVA water-based resin, foamed adhesive, painting lining, canvas restoration

## Abstract

The lining of paintings is a process of conservation science and art restoration used to strengthen, flatten, or consolidate paintings on canvas by attaching by means of adhesives a second canvas to the back of the existing one. To this aim, the prospects of the use of ethyl vinyl acetate (EVA) resins in aqueous dispersion applied as an adhesive in a foam form have been investigated in the present study. The key physical properties of the foam have been investigated, with a deep focus on rheological behavior and the drying rate, comparing the results with those obtained using the liquid products that are commercially available. Dedicated mock-ups have been prepared to test the adhesive for the lining process, inspecting adhesion strength, colorimetric properties, and the influence on the possible chromatic and visual alteration of the surface, also looking at the chemical interaction with painting materials and the deterioration after an artificial ageing process. The results obtained clearly indicated that the proposed technology is very suitable for the targeted application, and an EVA water-based foamed dispersion can be used for paintings’ lining, in view of the ease of application, being an appropriate adhesion, no chemical interaction, nor the deterioration of the painting.

## 1. Introduction and Research Objective

Lining paintings is a technique of rehabilitation of the textile support, which consists of the application on the back of the original painted canvas of a new canvas, which strengthens from the back the support of the pictorial film [1].

The common use of historical lining treatments in the 18th century was applying adhesives based on aqueous mixtures, such as animal glue, flour, resin, and Venice turpentine [2]. Afterward, during the 19th century, beeswax–resin mixtures came into use as lining adhesives in conjunction with hand ironing to press the two canvases together. Many improvements have been obtained toward hot-melt lining over the years, including the introduction of the hot table in 1946 and the further use of vacuum pressure (since 1955) in addition to heat [3]. In this way, the combined application of heat and moisture has a plasticizing effect on the paint and the ground, and the pressure flattens the distortions while the restraining effect of such stiff and strongly adhered support is thought to control their reappearance [4]. The first conference focused on re-lining treatments was held in 1974 during the International Symposium on Comparative Lining Techniques, at the Maritime Museum in Greenwich [5], during which new lining methods and materials were presented and the risks related to conventional glue paste and wax resin lining were advertised (i.e., darkening, adhesive penetration, or shrinkage of the fabric) [6]. Moreover, such a meeting laid the foundations for a new concept of restoration, based on reversibility and minimum intervention. During this international convention, Gustav Berger proposed his alternative solvent-based adhesive called BEVA 371, which is an ethylene vinyl acetate-based adhesive [7,8]. Such a substance is characterized by a large adhesive capacity, and it ensured the required protection against thermo-hygrometric variations [9,10]. Moreover, it guarantees the structural support to the painting, and it can maintain acceptable physic-chemical properties after aging [11,12,13].

Insofar, almost 50 years later, BEVA 371 is still one of the most widespread adhesives for 79% of restorers [6,14], despite its clear limitations and a very demanding application procedure. Indeed, this adhesive requires an activation step by means of heat, at a temperature between 40 and 70 °C, thus exposing the painting to a hot procedure that could be a cause of thermal stress. For this reason, over the last decade, different alternatives with no need for the hot activation step have gained interest, such as Beva Gel, PlextolB500, and paste glue, which are already available on the market, although they still need the application of pressure and could damage the painting both due to the stresses and yellowing during aging [15,16]. Moreover, as for BEVA 371, the evaporation of the solvent (i.e., toluene and naphtha in Beva Gel) during drying is hazardous both for the environment and the operator.

Therefore, the use of water emulsion-based adhesives has gained attention, since there are clear advantages related to minor toxicity for the absence of dangerous solvents and the absence of processing wastes. However, there is possible high penetration in the paintings of a large amount of water contained in the adhesive, due to its low viscosity, low boiling point, and its application on porous materials; this is the major limitation of the use of this kind of product, together with the high dependence on the environmental conditions, such as relative humidity and temperature [14,17].

The methodology proposed in this work is based on a different type of application of the adhesive that is modified in its consistency, from a moderately low viscous liquid into a high viscous foam, to be directly applied onto the lining fabric, and overlapped with the original canvas, with no need of heat activation. The foam is a type of dispersion in which a large proportion of gas by volume, in the form of gas bubbles, is dispersed in a liquid, solid, or gel, and the presence of a stabilizing agent (i.e., low-molecular-weight surfactants, polymers, proteins, nanoparticles, or their mixture) hinders the gas coalescence. Thus, the result is a network of gas/liquid interface, which provides the liquid foam with numerous and complex characteristics. The aim is to obtain an easy, rapid, economic, and user-friendly system to control the water-vertical diffusion, which represents one of the major problems in the use of water-based adhesives in lining procedures. The invention presented in this research [18] eliminates the use of hot activation as well as the use of hazardous solvents, leading to a significant simplification of the lining process by cutting off both the heat activation and the intermediate steps that the other adhesives required (i.e., the addition of extra layers, solvent spraying, use of pressure and temperature), as well as the need for personal protection equipment.

## 2. Materials and Methods

### 2.1. Materials

The complete list of the adhesives involved in this work are reported in Table 1. Below, the details of the various products are presented.

#### 2.1.1. EVA

EVA is an adhesive ethylene-vinyl acetate resins (E 43–63%; VA 37–57%) in aqueous dispersion, obtained with no addition of any plasticizer or surfactant. These products contain an emulsifier, based on polyvinyl alcohol (PVA or PVOH), to stabilize the water dispersion [19]. EVA was provided by C.T.S. srl (Firenze, Italy), under the trade name EVA ART.

#### 2.1.2. Beva Gel

BEVA^®^ gel is an aqueous dispersion of ethylene vinyl acetate and acrylic resins in a solution of water-soluble cellulosic material. It appears as a light brown gel. Beva Gel was provided by C.T.S. srl [20].

#### 2.1.3. Plextol B500

Plextol B500 is an aqueous dispersion of a thermoplastic acrylic polymer based on methyl methacrylate and ethylacrylate. Plextol B500 was provided by C.T.S. srl [21].

#### 2.1.4. Paste Glue

Paste glue was prepared by Zecchi Colori Belle Arti (Firenze, Italy) following the traditional Fiorentin recipe, based on rye flour, flax, wheat, strong glue, and molasses, and it requires low temperature activation [22].

#### 2.1.5. Origam

Origam canvas is a synthetic lining support that guarantees transparency, thanks to its single-yarn tissue (40 × 40 yarns/cm) composed of polyester fabrics. Origam was provided by C.T.S. srl.

#### 2.1.6. Ispra

Ispra canvas is a synthetic lining support; it is characterized by 15 × 15 yarns/cm polyester fabric composed of self-extinguishable fire-retardant material that guarantees dimensional stability and high lightfastness, moisture resistance, and tensile strength. Ispra was provided by C.T.S. srl.

#### 2.1.7. Canvas

Linen canvas for the support was provided by C.T.S. srl, while cotton canvas (Pieraccini) was provided by Amicucci materiali e colori per belle arti (Urbino, Italy).

#### 2.1.8. Paint Colors and Other Materials

C.T.S. srl provided the following items: Bologna Gypsum, in powder form, rabbit skin glue, MAIMERI linseed oil. Zecchi Colori Belle Arti provided English red earth, in powder form, and Siena earth, in powder form. Extra-fine acrylic colors titanium white and Cadmium Red were supplied by Iridron (Abralux Colori Beghè srl, Castelmarte, Italy).

**Table 1 polymers-15-01741-t001:** List of the different adhesives considered in this work (data were obtained from the product data sheet).

Adhesives	Solvent		Solid		References
Beva Gel	Nafta or toluene	60%	EVA copolymers cyclohexanone resin phthalate ester of hydroabietyl alcohol	40%	[20]
Plextol B500	Toluene	50%	Methyl methacrylate/ethyl acrylate copolymer	49–51%	[21]
Paste glue	Water	30%	Rye flour, flax, wheat, strong glue, molasses	70%	[22]
EVA	Water	50%	Ethylene-vinyl acetate resins	50%	[23]

#### 2.1.9. Foam Formation

The foam was produced in a liquid phase and was transformed into a liquid foam [24], introducing gaseous bubbles into the viscous water-based EVA emulsion. The foam was produced via a physical foaming mechanism, a liquid to gas transition mechanism, by means of a conventional steel whipped cream dispenser (WCD). The foaming agent was dinitrogen monoxide at 30 bar used in the ratio 32 g_N2O_/L_EVA_.

Foaming with WCD is a 3-step procedure: firstly, there is an injection of dinitrogen monoxide into the canister; then, shaking is required; and lastly, the application of pressure to the lever allow to dispense the foaming liquid as a foam [25].

The physical properties of the foam are evaluated in terms of foam quality [26]:(1)$$\Gamma \left[\%\right]=\left(\mathrm{gas}\;\mathrm{volume}/\mathrm{foam}\;\mathrm{volume}\right)\times 100$$

Four different WCDs were used to produce the foam for the sake of any kind of differences and to show that the method could be applied regardless of the device type used.

#### 2.1.10. Mock-Ups Preparation

Two different sets of specimens were prepared to evaluate EVA and foamed EVA performance as lining adhesives (Table 2).

The linen canvas (“A” series) was applied on an aluminum temporary canvas stretcher to easily spread the gypsum and rabbit glue preparation. Afterward, it was left attached to the frame for ten days to allow for stress-free drying. Later, oil paint layers were applied and left to dry for a few weeks.

The adhesives Plextol B500, liquid EVA, and paste glue were spread using a Teflon spatula on the lining canvases, Origam^®^ and Ispra^®^, placed on the back of the canvas support. The same procedure was followed for the foamed EVA-based resin by applying the product directly on the tissue. Afterward, lead weights were placed on the back of each canvas to prevent shifting or lifting, until completely dry. The procedure was slightly modified for Beva Gel, due to the need of a hot reactivation step, carried out by means of a vacuum hot table. The product was spread onto the canvas and placed on the table pre-heated to 40 °C, to ensure the evaporation of the solvent. The system was then sealed on a Melinex sheet, to allow the vacuum required for the complete bonding of the lining canvas to the support, and it was finally heated up to 60 °C and cooled down at room temperature slowly.

### 2.2. Adhesive Characterization

The experimental analysis on the liquid and foamed adhesives aimed to inspect the effect of foaming on the EVA resin, and to compare the properties with those of the different liquid adhesives that are commercially available, such as Plextol B500, Beva Gel, and paste glue. Moreover, the performance of the foam on the mock-ups was tested to simulate its application during the lining procedure.

#### 2.2.1. Drying

Gravimetric tests aimed to inspect the rate of evaporation of the volatile solvent from the adhesive, both from a film deposited ad hoc with a controlled thickness and spread on top of a relevant substrate. The procedure also allowed us to verify the actual amount of solvent present in the foam produced, and in the commercially available liquids. In all experiments, the different adhesives were applied on 5 cm diameter Petri dishes and dried under different values of relative humidity (RH = 0%, 45%, and uncontrolled room condition) until the sample weight was stable (measured every hour after the application), in order to evaluate the progressive weight loss until complete drying.

Furthermore, the adhesive foams produced were analyzed on the mock-up to inspect the adhesive drying rate on a system that simulated the targeted application.

#### 2.2.2. Density Tests

The density (ρ) of the foam was determined by filling a beaker with water and weighing it to obtain the exact volume of liquid inside (1 g of water corresponds to 1 cm^3^ of volume, as water density at 25 °C is about 0.997 g/cm^3^). By replenishing the beaker with the same volume of the adhesive in foam form and weighting it, the density of the foam was readily obtained and compared with the density of the adhesive in liquid form provided by its technical data sheet. The resulting density values may be used to calculate foam quality (Equation (1)).

#### 2.2.3. Rheological Behavior and Viscosity

The viscosity of the liquid adhesives (Beva Gel, Plextol B500, glue paste, and EVA) was determined via flow curve measurements with a stress-controlled rotational rheometer Anton Paar MCR 301 with Couette configuration. The analysis was carried out at 25 °C and the shear rate was increased logarithmically from 0.001 s^−1^ to 1000 s^−1^.

The flow curve analysis for the foam produced was carried out at 25 °C via rotational rheometry using a plate–plate configuration with a knurled surface to enhance the adhesion of the foam to the upper plate, applying a logarithmic shear rate ramp from 0.001 s^−1^ to 10 s^−1^ and by setting the gap between the two plates as being equal to 1 mm.

### 2.3. Mock-Up Characterization

#### 2.3.1. Adhesion Test

The effectiveness of the adhesives was tested by separating the two joined surfaces and evaluating the force exerted, thus determining the adhesive strength of different lining systems. Adhesion analysis was performed by means of the peel and lap shear tests, which follow normative ASTM D1002 [27] and ASTM F88 [28], respectively. The tests were carried out using an Instron tensile machine, equipped with a 10 kN load cell. In both cases, prior to each test, the specimens were kept under controlled relative humidity equal to 75% at room temperature for 2 weeks. The preparation of the test specimens was carried out according to the procedure prescribed by each standard norm. The peel test speed was set equal to 256 mm/s, while a value of 1.3 mm/s was considered for the lap shear tests. For each sample type, 5 repetitions were performed.

#### 2.3.2. Fourier Transform Infrared Spectroscopy (FTIR)

Fourier transform infrared spectroscopy (FTIR) analysis was carried out on cross-sections obtained from samples from the mock-ups. The samples were enclosed between two KBr pellets, embedded in a polyester resin, and finally polished. The analysis on the cross-sections was carried out using a Nicolet iNTM10MX microscope coupled with a mercury-cadmium-tellurium (MCT) detector, cooled with liquid nitrogen to reduce electrical noise of thermal origin. Attenuated total reflection (ATR) investigations on cross-sections were carried out using a conical germanium crystal of 300 μm diameter. The spectra were collected in a range from 4000 to 675 cm^−1^, with a spectral resolution of 4 cm^−1^ and 64 scans, with baseline correction and the smoothing of spectra. For each sample, a minimum of 3 to a maximum of 5 line maps were acquired along the whole stratigraphical structure, with measurements steps of 20 μm and 10 μm of spatial resolution.

#### 2.3.3. Penetration and Migration Analysis

An Olympus BX51M microscope, with oculars at 10x fixed magnitude and a set of objectives with 5×, 10×, 20×, 50×, and 100× magnifications, equipped with a filter U25LBD Olympus for the white light and a UV source Olympus U-RFL-T, was used for image acquisition, supported with PrimoPlus software connected to the microscope by means of the Olympus DP70 scanner. The Ponceau S (3-hydroxy-4-(2-sulfo-4-[4-sulfophenylazo] phenylazo)-2,7-naphthalenedisulfonic acid sodium salt), a sodium salt of a diazo dye of a deep red color, was employed as the contrast marker to evaluate the penetration and migration on the adhesive into the mock-ups.

#### 2.3.4. Colorimetry

A spectrophotometer and photoelectric colorimeter Minolta CM-2600d were used; the analysis was carried out in the spectral range from 400 to 700 nm, with three continuous scans, a small-lens-position (SAV) collimator with a diameter of 3 mm to increase resolution, and a D65 standard illuminant. For each specimen, 9 measurement points in the front side and 9 in the back side were chosen. The acquisitions were performed before the lining process, and then repeated shortly after it and after the application of the ageing protocol (Appendix A), comparing them with the samples without lining treatment. The measurements of the chromatic parameter referred to the CIE *L*a*b** chromaticity diagram and to the UNI 8941 standard [29]. The color difference Δ*E*^∗^_(*ab*)_ was calculated by the following equation [30]:(2)$$\Delta E_{\left(ab\right)}^{*}={\left[{\left(\Delta L^{*}\right)}^{2}+{\left(\Delta a^{*}\right)}^{2}+{\left(\Delta b^{*}\right)}^{2}\right]}^{\frac{1}{2}}$$

The thresholds were fixed to less than 3 as imperceptible alteration, between 3 and 6 as perceptible but not alarming, and over 6 as perceptible/highly perceptible with alarm [31].

The complete set of samples considered and the tests carried out are illustrated, for the sake of clarity, in the flow chart in Figure 1.

## 3. Results

### 3.1. Liquid and Foam Rheology

The rheological flow curves of the liquid commercial adhesives, compared to those obtained by the foamed EVA produced by four different WCDs, are shown in Figure 2. As expected, all of the behaviors observed were pseudoplastic, with a marked reduction in the viscosity as the shear rate applied increased. Moreover, no differences were detected in the rheological behavior of the foamed EVA samples produced by different WCD brands, which were proven to have no influence in the foam production process.

Table 3 reports the values of viscosity at shear rates of 0.1 s^−1^ and 10 s^−1^. Noteworthy is the fact that the foam produced cannot be too viscous, in order to maintain a good spread ability comparable to the solutions for lining paintings that are commercially available. Such a request was fulfilled thanks to the strong pseudoplastic behavior of the foam: the initial solid-like behavior at a very low shear rate switches into a liquid-like behavior once the shear is increased up to values larger than 0.1 s^−1^.

### 3.2. Foam Stability

The stability of the system, and the durability of the WCD filled with the EVA liquid emulsion and loaded with the foaming agent, were also inspected. To this aim, one dispenser was prepared at time zero and a foamed sample was produced: the rheological behavior was tested every week for 3 months, while keeping the EVA dispersion within the dispenser and storing it in environmental conditions, simulating a long shelf-life or an occasional use of the product. Figure 3 reports the resulting flow curves obtained at different times. As one can see, the rheological behavior remains constant over time. This is an important result since the durability of the foam is crucial both from an economical and an environmental point of view, and such a finding indicates the possibility of using the entire dispenser prepared, avoiding wastes of the product.

### 3.3. Drying

The analysis of the drying time provided the kinetics of water evaporation from either an adhesive thick layer or from a relevant support. Figure 4a shows the drying curve of a 1.2 cm thick layer of foam deposited onto a Petri dish, and it is noteworthy that the complete removal of the solvent (about 47 wt.%) that occurred within 24 h was not significantly affected by the environmental relative humidity.

Furthermore, 0.02 g/cm^2^ of foam was spread onto the surface of 5 cm diameter Origam and Ispra canvases, used as the lining canvases of a cotton one, in order to simulate the real application. As indicated in Figure 4b, only 3 h is required for the complete water evaporation, independently of the lining canvas used. Moreover, the weight loss at complete evaporation is 47%, as for the Petri dish drying test, in line with the product specification. Therefore, no water can be trapped in the solidified external regions of the foamed adhesive, thus avoiding possible water vertical diffusion phenomena at longer times.

The drying behavior of the other commercial glues (Beva Gel, Plextol B500, glue paste, and Liquid EVA water-based emulsion) was tested for the sake of comparison, to inspect the drying time of a 0.8 cm thick layer spread on a Petri dish (Figure 5). Although the first three systems were the most common adhesives for lining paintings, it was found that the time required to the complete evaporation was four times longer with respect to the foamed EVA. Moreover, the long time needed for the evaporation of the solvent led to the formation of a solidified skin that slowed down the evaporation process, and part of the solvent remained trapped. Such an effect when occurring between the painting and the canvas may lead to permanent damages to the painting. Table 4 (and Figure 5) highlights that the use of a foamed adhesive can shorten drying time, and it prevents any possible formation of solid and less permeable skin, which may hinder solvent evaporation. These aspects are crucial not only from the technical/technological point of view, but since the solvent (i.e., toluene for Beva Gel) is often toxic in most of the commercial adhesives in the field, safety concerns may be associated with the adhesive application.

Furthermore, an additional advantage of the foamed solution is related to the amount of adhesive needed to obtain the total coverage of the surface and thus to ensure the complete adhesion of the lining canvas. The amount of the foamed EVA, indeed, is always significantly lower than that required by the liquid EVA resin. Such differences are reported in Table 5 and the results are also influenced by the material considered for the preparation of mock-ups.

### 3.4. Peel and Lap Shear

The mechanical tests carried out on the mock-ups allowed us to compare the adhesion strength of the foamed EVA adhesive with the other commercial solutions. It is useful to remind readers that all samples tested were fabricated considering the same canvas, using the prescribed method for re-lining, and spreading the adhesives by means of a spatula. Therefore, the amount of adhesive deposited may vary (see, e.g., Table 5), as well as the thickness of the bonding layer. However, such differences are not expected to significantly affect the adhesion tests.

Table 6 highlights the results obtained using either Origam or Ispra as the lining canvas. Relevantly, when Origam was used in the mock-up preparation, the failure of the canvas occurred before the end of the adhesion tests for all of the samples, except for foamed EVA and paste glue. Therefore, the comparison with the optimal performances in terms of adhesive capacity and adhesive strength indicated in the technical literature (280–400 N/m for peeling and to 0.24 MPa for shear [32]) was performed on Ispra canvas samples only. Plextol B500, foamed Evart, and Beva Gel provide values close to the optimal ones, but it is noteworthy that during the tests, the use of both Plextol and Beva Gel led to the detachment of the layer that simulated the painting with the Ispra canvas, although the adhesion strength results were quite large. Conversely, foamed EVA shows satisfactory results, without damaging the canvas to any extent; the adhesion failure occurs right within the adhesive layer, with no possible consequences for the painting.

### 3.5. Surface and Color Alteration

The results of colorimetric analysis on the mock-ups front side (Table 7, Figure 6 and Figure 7) showed a similar chromatic alteration due to a post degradation process on the specimens with and without adhesive application on the two lining canvases, with Δ*E*^∗^_(*ab*)_ values always remaining well below the alert threshold (Δ*E*^∗^_(*ab*)_ > 6). Concerning the “M” series (Ispra support), the Δ*E*^∗^_(*ab*)_ values were always kept below 3 (i.e., the limit of perceptibility), with a lower value for the foamed adhesive (Δ*E*^∗^_(*ab*)_ foamed EVA = 1.88; Δ*E*^∗^_(*ab*)_ liquid EVA = 2.82). Regarding the samples supported by Origam, Δ*E*^∗^_(*ab*)_ rose above the value of 3 (alteration defined as perceptible but not alarming), with no significant differences between the two forms of resin application (Δ*E*^∗^_(*ab*)_ foamed EVA = 3.11; Δ*E*^∗^_(*ab*)_ liquid EVA = 3.12). Observing the spectral curves, a slight decrease in the reflectance is apparent in the range between 600 and 740 nm for all of the specimens of the “M” series after the degradation process, compared to the original color. In the “A” series, the Δ*E*^∗^_(*ab*)_ values with Ispra support rose above the value of 3 (Δ*E*^∗^_(*ab*)_ foamed EVA = 3.93; Δ*E*^∗^_(*ab*)_ liquid EVA = 4.19), while with the Origam as the lining canvas, Δ*E*^∗^_(*ab*)_ was lower than 3 in the case of EVA resin applied in foamed form and higher than the limit of perceptibility in liquid one (Δ*E*^∗^_(*ab*)_ foamed EVA = 2.58; Δ*E*^∗^_(*ab*)_ liquid EVA = 5.54). The spectral reflectance curves show, for both forms of resin applications, a significant decrease in reflectance, especially in the 570–620 nm range, and more highlighted in the 620–740 nm range, as a consequence of a strong darkening of the specimen surface after the degradation process. However, the same surface darkening also occurs in specimens without the EVA resin application, as also demonstrated by Δ*E*^∗^_(*ab*)_ values (Table 7 and Table 8); therefore, the alteration can mainly be ascribed to the natural ageing of oil paint layers.

### 3.6. Penetration of the Adhesive

The relined canvas treated with Pounceau S dye was cut and the resulting cross-sections were examined using an optical microscope. Figure 8 shows the diffusion and the extent of penetration of the adhesive through the painting stratigraphy. The specimens fabricated with the foamed EVA resin show the marker visible across the original and the lining support layers, while no penetration was observed inside the ground layer. On the contrary, the preparation layer resulted moderately pink and partially spotted due to the application of liquid EVA resin, which confirms its unsuitability for such an application in its original form (liquid water emulsion).

Such observations are also validated by the FTIR analysis on the preparation layer, as illustrated in Figure 9, which reports the spectra collected on the preparatory ground layer (the exact spots are indicated in the optical images also present in the figure). The IR spectra are characterized by a predominance of absorption bands related to calcium sulphate dihydrate (strong S-O stretching in the region 1200–1050 cm^−1^), two sharp peaks at 1620 and 1680 cm^−1^ due to O-H bending and O-H stretching in region 3600 and 3200 cm^−1^ [33], related to proteins with amide I, amide II, and amide III, and less intense absorption bands near 1650, 1550, and 1450 cm^−1^, respectively [33], due to the animal glue as a binder for gypsum. No vibration bands related to the water-based EVA resin [34,35] were detected on the samples with the foamed product applied, confirming that no significant penetration occurs in the ground layers above the canvas. Conversely, linear maps on samples prepared with liquid EVA water-based adhesive show, along the thickness of the cross-section, the continuous presence of the carbonyl band at 1735 cm^−1^, the ester C-O stretching at 1240 cm^−1^ and 1022 cm^−1^ as the small shoulder, and the methyl C-H rocking at 795 cm^−1^ [34,35], which are attributable to the ethylene-vinyl acetate resin. A gradual decrease in the intensity of such absorption bands is observed until they disappear altogether with the beginning of the paint layers.

Such findings have to be ascribed to the larger drying velocity of the EVA resin when foamed (thanks to its porosity), which prevents the adhesive penetration into the painting: the faster the drying process is, the less the penetration in the substrate. Similarly, water can be readily removed from the lining layer, thus not affecting to any extent the painting. This is also supported by the rheological tests, indicating that the foamed product has a higher viscosity at small shear rates than the liquid one, and it drops only at high stress values, as during the drawing up, thus allowing one to obtain a good spread ability of the adhesive. Such a larger viscosity, coupled with the increase in volume that allows the use of smaller amounts of resin to be obtained, but a uniform coverage of the same surface, enables the prevention of the penetration of the adhesive and of the water along the stratigraphy of the mock-ups.

The obtained result is a key factor, as it influences not only the interactions between the adhesive and the materials upon which it is applied, thus preserving the integrity of the painting, but also the invasiveness and successive reversibility of the lining operations. In fact, the amount of product that could remain on the support of a painting after a de-lining procedure and the degree of penetration, as well as the possibility of removing it in a non-invasive way, are other key factors that support the use of this solution for such an application.

The micro-FTIR analysis revealed that no appreciable aging can be observed on the samples subjected to artificial degradation, as no footprint of EVA resin degradation products was observed in appreciable quantities according to the technique used. The IR spectra, in fact, show neither absorption bands in the 1700–1720 cm^−1^ region, relating to the formation of ketones, nor at 1785 cm^−1^ for lactone formation, resulting from the typical hydrolysis and acetic acid formation by the de-acetylation of the vinyl fractions present in the EVA polymer. No decrease in intensity of the carbonyl band at 1735 cm^−1^ was registered [36,37]. The absence of such components supports the stability of the EVA water-based adhesive and its suitability to the targeted application.

## 4. Conclusions

The solution proposed in this work represents a new frontier for lining paintings and its results show that it will be competitive in the market, not only for its excellent physico-chemical properties but also from the environmental and safety points of view.

Similarly to commercially available adhesives, foamed EVA water-based resin maintains the pseudoplasticity for the low values of shear rates considered, analogous to conventional adhesives in the application process. Moreover, foamed EVA overcomes the issue of water vertical diffusion, since the rate of drying is so fast that it is impossible for water penetration to occur inside the painting. Such a behavior avoids the formation of the gel layer which traps the solvent leading to the damage of the painting. The resulting adhesion properties are adequate for the application and in line with the other products for paintings’ relining.

Furthermore, the use of water-based resin does not require the use of any safety device for the operators, making it different to most of the commercially available adhesive solutions now available on the market. Unlike commercial glues, foamed EVA-based resins do not yellow with time and they maintain their adhesive strength without ruining the painting; in addition, the lining with the foam guarantees a transparent lining which gives an insight to the quality of the adhesion and preserves the visibility of historical data.

Therefore, the results of the experimental campaign showed that the formulation of foamed EVA, obtained via a simple foaming process of EVA water-based resins to conveniently modify its viscosity, could represent a valid alternative to conventional systems, also suitable for other more complex and laborious lining systems. Differently to traditional adhesives that currently dominate the scene, this application method is able to provide satisfactory results with no use of organic solvents, which may be hazardous or toxic, and no heat-seal activation required to guarantee the adequate and desired mechanical properties.

## Figures and Tables

**Figure 1 polymers-15-01741-f001:**
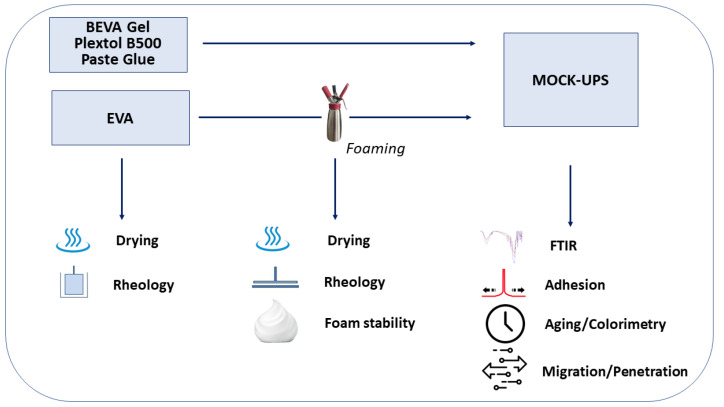
Experimental flow chart.

**Figure 2 polymers-15-01741-f002:**
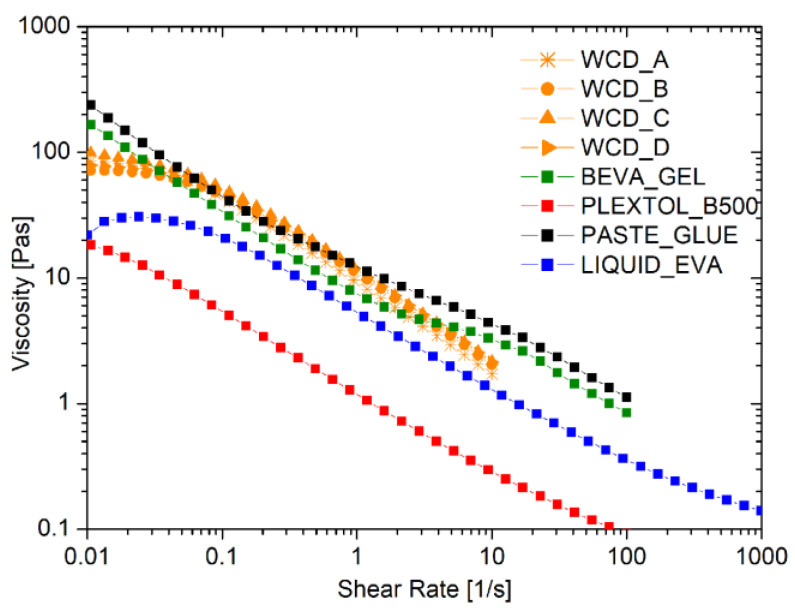
Flow curves of commercial adhesives, liquid, and foamed EVA using different devices for the formation of the foam, namely whipped cream dispensers (WCDs) A, B, C, and D.

**Figure 3 polymers-15-01741-f003:**
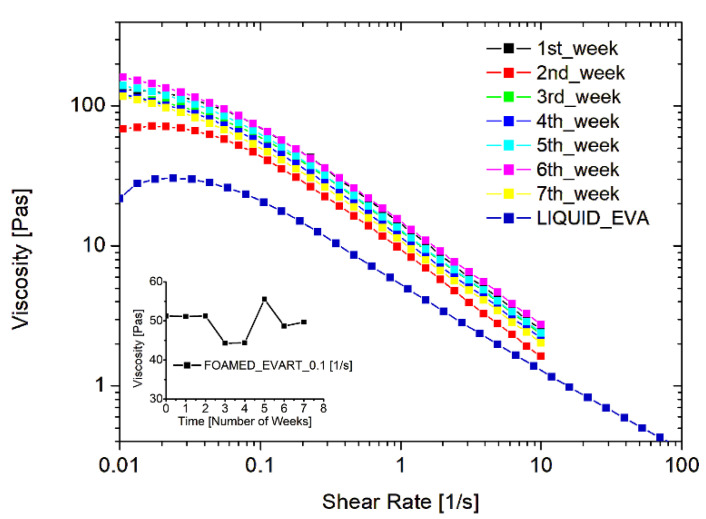
Stability of the foamed EVA adhesive, filled in the WCD, in terms of rheological behavior of the foam just produced.

**Figure 4 polymers-15-01741-f004:**
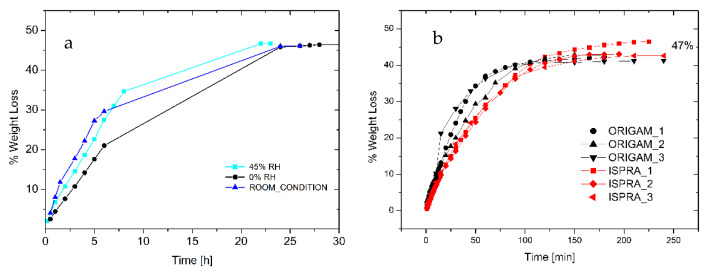
(**a**) Drying curve of foamed EVA adhesive under different environmental RH (adhesive layer thickness 1.2 cm). (**b**) Drying of foamed EVA adhesive onto linen canvas relined with Origam and Ispra.

**Figure 5 polymers-15-01741-f005:**
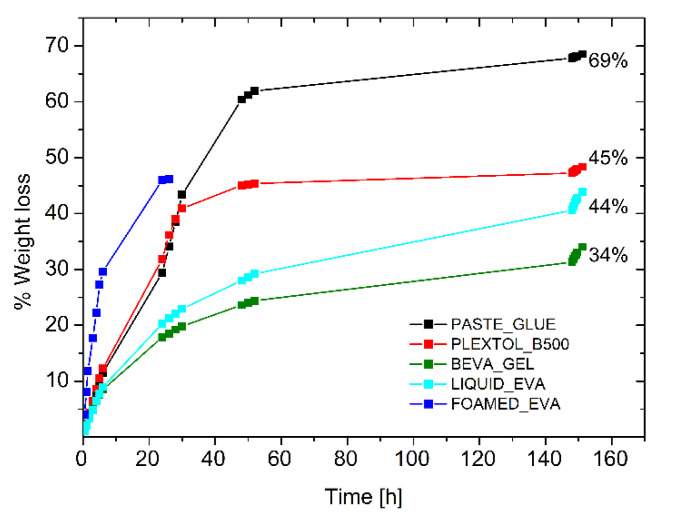
Drying time of commercial adhesives and foamed EVA.

**Figure 6 polymers-15-01741-f006:**
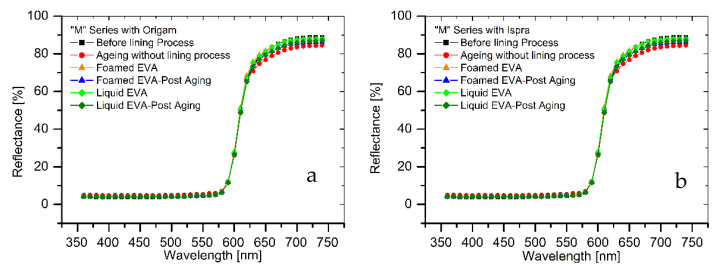
(**a**) Spectral reflectance curves acquired on “M” series specimens with Origam. (**b**) Spectral reflectance curves acquired on “M” series specimens with Ispra.

**Figure 7 polymers-15-01741-f007:**
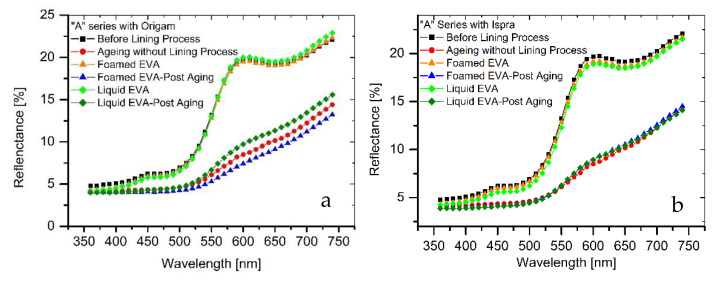
(**a**) Spectral reflectance curves acquired on “A” series specimens with Origam; (**b**) spectral reflectance curves acquired on “A” series specimens with Ispra.

**Figure 8 polymers-15-01741-f008:**
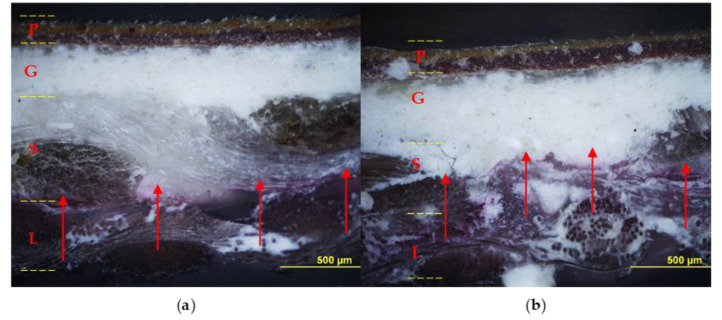
Cross-sections: on the left sample lined with foamy adhesive, on the right sample lined with liquid adhesive. Optical microscope photographs in visible light (**a**,**b**) and UV light (**c**,**d**) with contrast marker applied: pictorial layer (P), ground layer (G), original canvas (*S*), and canvas lining (*L*). The EVA-based adhesive in foamed form (**a**,**c**) penetrated to the junction between the original canvas (*S*) and the lining canvas (*L*); instead, the adhesive without the foaming process (**b**,**d**) also penetrated the ground layer (*G*).

**Figure 9 polymers-15-01741-f009:**
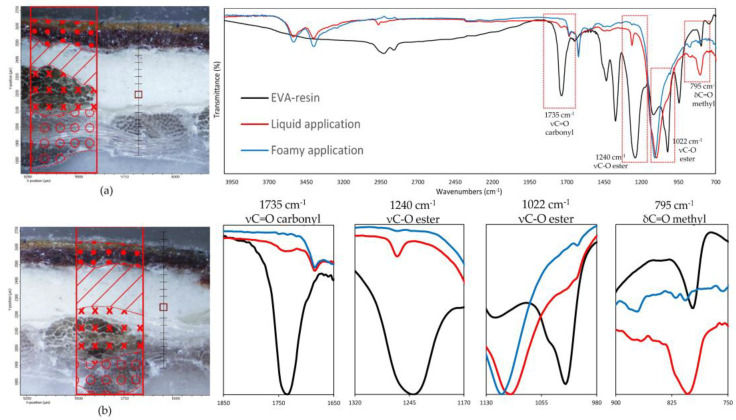
(**On the left**) FT-IR line maps on cross-sections: (**a**) foam EVA-based adhesive application, (**b**) liquid EVA-based adhesive application; painting layers (dots); preparatory layer (diagonal lines); original canvas (crosses); lining canvas (circles); measurement spots collected in the preparatory layers (red square). (**On the right**) FT-IR spectra on the measurement spots (red square); (blue line) foam EVA-based adhesive application; (red line) liquid EVA-based adhesive application. (Black line) EVA-resin standard. In the red spectrum, the vibrational bands related to the EVA-based resin characteristic C=O carbonyl (1735 cm^−1^), C-O ester (1240, 1022 cm^−1^), and C-H methyl (795 cm^−1^) functional groups are clearly visible.

**Table 2 polymers-15-01741-t002:** Structure of mock-ups prepared, and materials used.

Layers	“A” Series	“M” Series
Paint	English red earth in oil medium Siena earth in oil medium	Cadmium red in acrylic medium Titanium white in acrylic medium
Preparation	Gypsum and animal glue	Acrylic titanium white primer
Support	Linen canvas (8 × 10 yarns/cm^2^)	Cotton canvas (16 × 10 yarns/cm^2^)

**Table 3 polymers-15-01741-t003:** Viscosity obtained via rheological tests on the different adhesives, evaluated at certain shear rates.

Adhesives	Viscosity	
	$\dot{\gamma }=0.1\;s^{-1}$	$\dot{\gamma }=10\;s^{-1}$
Beva Gel	31.1	2.70
Plextol B500	5.02	0.27
Paste glue	40.9	4.12
Liquid Evart	20.5	1.27
Foamed EVA (WCD A)	54. 6	2.34
Foamed EVA (WCD B)	46.0	2.01
Foamed EVA (WCD C)	50.1	2.13
Foamed EVA (WCD D)	49.3	2.16

**Table 4 polymers-15-01741-t004:** Drying time of commercial adhesives and foamed EVA.

Adhesives	Drying Time (Thick Layer)
Beva Gel	350 h
Plextol B500	350 h
Paste Glue	160 h
Evart	350 h
Foamed EVA	24 h

**Table 5 polymers-15-01741-t005:** Amount of EVA adhesive, in foam and liquid form, applied in each series of specimens (mg/mm^2^).

	“A” Series		“M” Series	
	Ispra^®^	Origam^®^	Ispra^®^	Origam^®^
Foamed EVA	0.154	0.123	0.242	0.165
Liquid EVA	0.292	0.200	0.304	0.188

**Table 6 polymers-15-01741-t006:** Lap shear strength and peel strength of adhesives.

Adhesive	Canvas	Lap Shear Strength	Peel Strength
		MPa	N/m
BEVA 371	Origam^®^	0.115 ± 0.005	63.0 ± 12.8
Plextol B500		0.073 ± 0.10	387 ± 183
Foamed EVA		0.10 ± 0.02	135 ± 105
Beva Gel		0.12 ± 0.007	341 ± 174
Paste glue		0.014 ± 0.004	3.29 ± 1.54
BEVA 371	Ispra ^®^	0.11 ± 0.03	35.2 ± 3.6
Plextol B500		0.40 ± 0.014	218 ± 65
Foamed EVA		0.30 ± 0.02	122 ± 24
Beva Gel		0.34 ± 0.04	160 ± 118
Paste glue		0.019 ± 0.015	4.58 ± 0.93

**Table 7 polymers-15-01741-t007:** Comparison between the differences in the colorimetric parameters of foamed EVA resin and liquid EVA resin artificially degraded mock-ups and the colorimetric parameters of artificially degraded mock-ups without EVA resin.

Series	Lining Support	Application Method	Δ*L*^∗^	Δ*a*^∗^	Δ*b*^∗^	Δ*E*^∗^_(*ab*)_
*M*	Ispra^®^	Foamed EVA resin	−1.14	0.78	1.28	1.88
		Liquid EVA resin	−0.43	1.87	2.07	2.82
	Origam^®^	Foamed EVA resin	−0.09	2.11	2.28	3.11
		Liquid EVA resin	−0.61	1.99	2.33	3.12
*A*	Ispra^®^	Foamed EVA resin	0.93	2.94	2.44	3.93
		Liquid EVA resin	0.84	2.94	2.86	4.19
	Origam^®^	Foamed EVA resin	−1.35	2.04	−0.83	2.58
		Liquid EVA resin	1.96	3.56	3.77	5.54

**Table 8 polymers-15-01741-t008:** Series “M”. Colorimetric parameters recorded on mock-ups without the application of any adhesive and before the artificial degradation process, compared with the colorimetric parameters recorded after the artificial degradation, both on mock-ups with foamed EVA resin and liquid EVA resin and without adhesive application.

	Application Method	*L* ^∗^	*a* ^∗^	*b* ^∗^	Δ*L*^∗^	Δ*a*^∗^	Δ*b*^∗^	Δ*E*^∗^_(*ab*)_
Before lining process		43.32	12.70	24.33				
Ageing without lining process		30.06	5.78	9	−13.26	−6.92	−15.33	21.42
Ispra^®^	Foamed EVA resin	30.99	8.72	11.44	−12.33	−3.98	−12.89	18.28
	Liquid EVA resin	30.9	8.72	11.86	−12.42	−3.98	−12.47	18.04
Origam^®^	Foamed EVA resin	28.71	7.82	8.17	−14.61	−4.88	−16.16	22.33
	Liquid EVA resin	32.02	9.34	12.77	−11.30	−3.36	−11.56	16.51

## Data Availability

Not applicable.

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
