# Peer review of "Liquid Foam-Ethyl Vinyl Acetate Adhesive Systems for Lining Process of Paintings: Prospects of a User-Friendly, Harmless Alternative to Conventional Products"

_polymers, 2023, doi:10.3390/polym15071741_

Round 1
Reviewer 1 Report
1. Line no. 21- "looking also to"replace with " also looking into"
2. Check Text alignment at many places are not proper "make it justify alignment "
3. All the figures are of poor quality.
4. Check the line space/ word space (for example- line 361-361)
5. Insufficient recent literature is presented to support the aim of the study. This point still needs further revision.
6. There is no quantitative discussion on results in conclusion, needs to be revised.
7. Methodology of the proposed model must be illustrated by a clear flowchart.
Author Response
Extensive editing of English language and style required.
-->The manuscript has been revised thoroughly, aiming to improve the use of English language and style .
Line no. 21- "looking also to" replace with " also looking into"
-->The sentence has been revised according to the referee’s suggestion.
Check Text alignment at many places are not proper "make it justify alignment "
-->The layout of the text has been improved.
All the figures are of poor quality.
-->The figure quality has been checked and improved, in order to be clear in the pdf format.
Check the line space/word space (for example- line 361-361)
-->The layout of the text has been improved.
Insufficient recent literature is presented to support the aim of the study. This point still needs further revision.
-->The introductory section has been expanded, according to the referee’s suggestion and reference to newer papers has been added.
There is no quantitative discussion on results in conclusion, needs to be revised.
-->The conclusion section has been improved in order to meet the referee’s suggestion. In particular, the authors decided to merge the Discussion and Conclusion section.
Methodology of the proposed model must be illustrated by a clear flowchart.
-->A new figure with a clear flowchart of the samples prepared and tests executed has been added, thanking the referee for the great suggestion.
Reviewer 2 Report
The authors present ethyl vinyl acetate foam (EVA foam) for paining lining process. They compared EVA foam with commonly used adhesives for lining in aspects of adhesion, colorimetry, and water penetration. The EVA foam was applied in environmentally-friendly solvent water, and shows good stability, fast drying, good adhesion and low penetration to the other layers. Overall, this study provides comprehensive comparisons of various adhesive materials for lining process and demonstrates the advantage of EVA foam. This manuscript is suitable to publish after addressing the following issues.
1. line 72, what is (ref. patent)?
2. All figures are very blurred. Please replace them with high-resolution figures.
3. Table 1 and 6. Does "Evart" mean "EVA"? "Beva" means "BEVA"? Please keep consistent.
4. For the stability test, under what conditions was the foam kept for 3 months? Except for the viscosity test, the stability can also be evaluated by the adhesive performance after 3 month.
5. For adhesion comparison, did all samples have the same thickness?
6. For the FTIR test of cross-section, which position of the ground layer was tested? The borders look vague. Could the authors do an FTIR mapping throughout the cross-sectionto show the penetration of adhesive materials?
Author Response
line 72, what is (ref. patent)?
-->The reference link was not correctly displayed; the sentence has been now corrected, and it contains the reference to the patented work.
All figures are very blurred. Please replace them with high-resolution figures.
-->Such undesired effect, related to the pdf generation of high quality images, has been overcome changing the format of the images and the importing method.
Table 1 and 6.Does "Evart" mean "EVA"? "Beva" means "BEVA"? Please keep consistent.
-->The nomenclature has been homogenized throughout the whole manuscript, according to the referee’s suggestion.
For the stability test, under what conditions was the foam kept for 3 months? Except for the viscosity test, the stability can also be evaluated by the adhesive performance after 3 months.
-->Once the liquid resin was poured into the WCD, it was stored within the disperser at room conditions, simulating a simulating a long shelf life or even an occasional use of the product. Such information is now clearly included in the manuscript for the sake of clarity.
For adhesion comparison, did all samples have the same thickness?
-->The samples tested have been fabricated considering the same canvas and using the prescribed method for re-lining (by the adhesive producers or distributors), and spreading the adhesives by means of a spatula. Therefore, the amount of adhesive deposited may vary, as well as the thickness of the bonding layer. However, such differences are not expected to affect significantly the adhesion tests. Furthermore, such tests have been performed on mock-ups that are simulating real systems. The text has been modified in order to clarify this point.
For the FTIR test of cross-section, which position of the ground layer was tested? The borders look vague. Could the authors do an FTIR mapping throughout the cross-section to show the penetration of adhesive materials?
-->The FTIR spectra are collected on the preparatory ground layer to inspect any possible penetration of the EVA based resin into the canvas that may lead to painting damaging. To prevent any misunderstanding, the exact spots in which the analysis was carried out are indicated in the optical images also present in the figure, which illustrates the different layers.